# Adaptive Labeling for Efficient Out-of-distribution Model Evaluation

**Daksh Mittal,**[*] **Yuanzhe Ma,**[*] **Shalmali Joshi,  Hongseok Namkoong**
Columbia University
{dm3766, ym2865, sj3261, hn2369}@columbia.edu

## Abstract

Datasets often suffer severe selection bias; clinical labels are only available on patients for whom doctors ordered medical exams. To assess model performance outside the support of available data, we present a computational framework for adaptive labeling, providing cost-efficient model evaluations under severe distribution shifts. We formulate the problem as a Markov Decision Process over states defined by posterior beliefs on model performance. Each batch of new labels incurs a "state transition" to sharper beliefs, and we choose batches to minimize uncertainty on model performance at the end of the label collection process. Instead of relying on high-variance REINFORCE policy gradient estimators that do not scale, our adaptive labeling policy is optimized using path-wise policy gradients computed by auto-differentiating through simulated roll-outs. Our framework is agnostic to different uncertainty quantification approaches and highlights the virtue of planning in adaptive labeling. On synthetic and real datasets, we empirically demonstrate even a one-step lookahead policy substantially outperforms active learning-inspired heuristics.

## 1   Introduction

Engineering progress in AI is predicated on rigorous empirical evaluation. Evaluating supervised ML models is easy when there is no distribution shift. However, naive offline evaluations on observational data cannot reliably assess safety due to selection bias. When ground-truth outcomes are expensive to collect, existing supervised datasets are often heavily affected by selection bias. For example, AI-based medical diagnosis systems are typically evaluated using past clinical labels, which are only available for patients who were initially screened by doctors. This creates a fundamental limitation: we cannot even assess the model's performance on patients who were never screened in the first

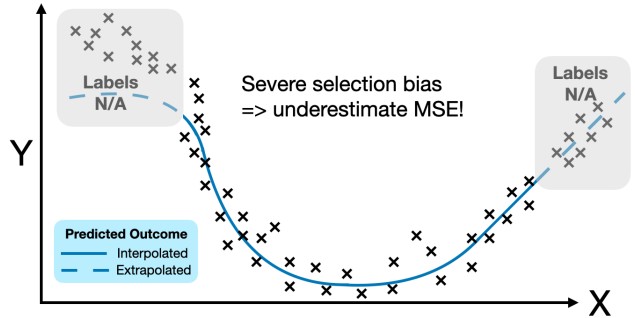

Figure 1: **Adaptive labeling to reduce epistemic uncertainty over model performance.** Among the two clusters of unlabeled examples (left vs. right), we must learn to prioritize labeling inputs from the left cluster to better evaluate mean squared error.

place. Moreover, due to the absence of positive pathological patterns in the available data, previously unseen patient types may never get diagnosed by the AI diagnosis system. Evaluating the model's performance on previously unscreened patients is a first step toward mitigating this negative feedback loop.

---

[*]Equal contribution

38th Conference on Neural Information Processing Systems (NeurIPS 2024).

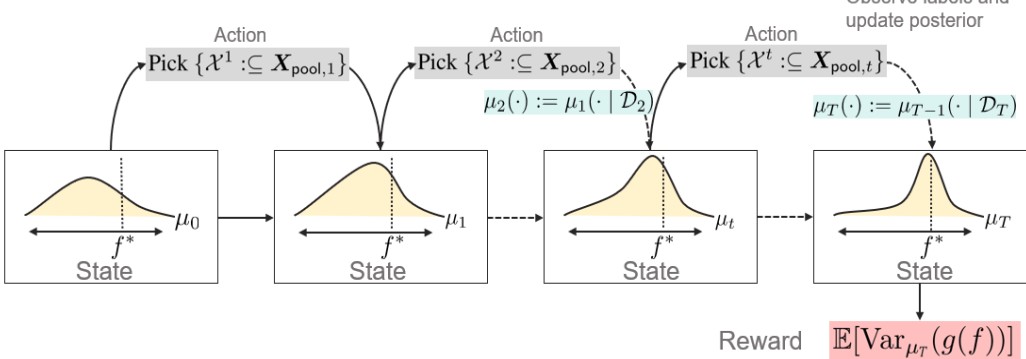

Figure 2: **Overview of our adaptive sampling framework.** At each period, we select batch of inputs $X^t$ to be labeled, and obtain a new labeled data $\mathcal{D}_t$. We view posterior beliefs $\mu_t(\cdot)$ on $f^\star(Y|X)$ as "states", and update it as additional labeled data is collected. Our goal is to minimize uncertainty on the performance of the prediction model $\psi(\cdot)$ at the end of $T$ periods.

In this paper, we study model evaluations under severe distribution shifts, where the support of the available supervised data is different from examples encountered during deployment. Randomly selecting inputs $X$ to collect additional outcomes/labels $Y$ incurs prohibitive cost, especially when the outcome distribution varies over a high-dimensional feature space. To address this, we must focus our sampling efforts on out-of-distribution (OOD) examples where the model's performance is uncertain. Going beyond the OOD detection literature that focuses on the distribution of input $X$ [41], we further connect the rarity of $X$ with how smooth we believe $Y|X$ to be. For example, in the extreme case where we believe $Y|X$ is linear, it is easy to extrapolate model performance to regions of $X$ where no data is available; such extrapolation is more challenging for nonlinear models.

We propose an adaptive sampling framework that focuses sampling effort on inputs $X$ where there is large *epistemic* (actionable) uncertainty on the shape of $Y|X$. We differentiate this with *aleatoric* (irreducible) uncertainty, which is due to idiosyncratic noise in the outcome measurement process. Since updating the sampling policy requires engineering or organizational effort in practice, we consider a few-horizon setting where at each period, a *batch* of inputs is selected for labeling.

Our main contribution is the formulation of a *planning problem* for selecting batches of inputs from a large pool of unlabeled data (Section 3). We develop adaptive policies that incorporate how beliefs on model performance get sharper as more labels are collected. We model our current belief on the data generation process $Y|X$ as the current state. After observing a new batch of labeled data, we update our belief on the epistemic uncertainty via a posterior update ("state transition"). Our goal is to minimize the uncertainty over model performance at the end of label collection, as measured by the variance under the posterior after the final batch is observed. Modeling each batch as a time period, we arrive at a Markov Decision Process (MDP) where states are posterior beliefs, and actions correspond to selecting subsets (batches) from the pool of examples to be labeled (Figure 2). Notably, our problem is less ambitious than an active learning problem [34] where the goal is to adaptively label data to *improve* model performance, over a *large* number of batches. Narrowing our focus to model *evaluation* allows us to address a broad class of evaluation problems including regression models, and solve the problem more effectively.

Solving our planning problem is computationally challenging due to the combinatorial action space and continuous states. To address this, we develop a tractable approach by considering a smooth relaxation of the problem, where we optimize over smoothly parameterized policies that select an entire batch of inputs [39]. Although the well-known "score trick" enables the computation of policy gradient estimates based on function evaluations alone ("REINFORCE") [36, 5], this method suffers from high variance and may lead to poor performance in practice.

To compute reliable policy gradients, we develop an *auto-differentiable* planning framework in Section 4. Since the dynamics of the MDP (posterior updates) are known, we perform "roll-outs" using the current posterior beliefs: we simulate potential outcomes that we may obtain if we label a particular batch, and evaluate the updated uncertainty on model performance based on the imagined pseudo labels of this new batch. By implementing all the operations within an auto-differentiable framework, we can backpropagate the observed reward to the policy parameters and calculate the

*pathwise* policy gradient. While our original problem is nonsmooth and discrete, auto-differentiating over this smoothed MDP provides reliable policy optimization methods.

Our approach is agnostic to the uncertainty quantification approach, and allows incorporating latest advances in Bayesian modeling (e.g., [5, 28]). Our "model-based" approach rests on encoding state transitions (posterior updates) in an auto-differentiation framework. We stress that we do not require the posterior updates to have a closed form (conjugacy), and allow approximate posterior inference using gradient-based optimizers. We demonstrate our planning framework over beliefs formed by Gaussian processes, and Deep Ensembles [18, 26] that enable leveraging the inductive biases of neural networks in high dimensions and gradient descent methods to update probabilistic beliefs over $Y|X$. Empirically, we observe that even a single planning step can yield significant practical benefits (Section 5). On both real and simulated datasets with severe selection bias, we demonstrate that one-step lookahead policies based on our auto-differentiation framework achieve state-of-the-art performance, outperforming REINFORCE policy gradients and active learning-based heuristics.

**Limitations** Our empirical validation highlights several open research directions required to scale our framework. First, although recent advances in Bayesian modeling have achieved substantial progress in one-shot uncertainty quantification, we empirically observe that it is often difficult to maintain accurate beliefs as more data are collected. Second, we detail engineering challenges in implementing our auto-differentiation framework over multi-period roll-outs, highlighting the need for efficient implementations of high-order auto-differentiation.

## 2 Related work

Our work is closely related to active learning [1, 34], where the modeler adaptively collects labels to *improve* model performance. Active learning methods typically focus on classification models and select inputs to label based on uncertainty sampling techniques such as margin-sampling or entropy, or by leveraging disagreements among different models, exemplified by query-by-committee and Bayesian Active Learning by Disagreement (BALD) [14]. Although there is no notion of planning, some query strategies account for the expected reduction in error or model change [34]. Since it is important to collect labels over a diverse set of inputs, heuristic criteria based on diversity and density are sometimes incorporated to account for the feature space distribution [34]. Prior work addressing distribution shifts in active learning [42] uses importance weighting, which is not feasible in our setting, where the distribution shift is out of support. Batched settings are a long-standing challenge in active learning [34]. In Section 5, we show that adapting greedy-based heuristics to batched scenarios yields poor performance. Recent extensions of active learning to batched settings (e.g., BatchBALD [17], BatchMIG [37]) continue to rely on myopic greedy algorithms for classification. On the other hand, we leverage our limited scope on evaluation to formalize a planning problem and derive a unified computational framework that can handle regression problems. We provide lookahead policies [2, 6, 7] that plan for the future and flexibly handle different objectives and uncertainty quantification methodologies.

Our problem can also be viewed as a version of pure exploration bandits [23] where we want to minimize some given function of the posterior. In contrast to conventional formulations, our central focus on batching induces combinatorial action spaces and few horizons. Bayesian optimization methods optimize black-box functions that are computationally challenging to evaluate by modeling the function as a draw from a Gaussian process [8]. Notably, the knowledge gradient algorithm [9] maximizes the single period expected increase in the black-box function value and can be viewed as a one-step lookahead policy. Using auto-differentiation methods, we extend these classical ideas to model evaluation by incorporating batching (combinatorial action spaces).

While Gaussian processes works well in low dimensions, quantifying epistemic uncertainty on $f^\star(Y|X)$ for high dimensional inputs is an active area of research, including popular techniques such as dropout [10], Bayes by Backprop [3], Ensembles/ Ensemble+ [18, 26], and Epistemic Neural Networks [28]. Our framework is compatible with deep learning-based uncertainty quantification models, such as Ensembles, where we can only do approximate posterior updates.

## 3 Markov decision process over posterior beliefs

Our goal is to evaluate the performance of the model $\Psi : \mathcal{X} \to \mathcal{Y}$ over the input distribution $P_X$ that we expect to see during deployment. Given features $X \in \mathcal{X}$, outcomes/ labels are generated from

some unknown function $f^\star\colon Y = f^\star(X) + \varepsilon$ where $\varepsilon \sim N(0, \sigma^2)$. When ground truth outcomes are costly to obtain, previously collected labeled data $\mathcal{D}^0$ typically suffers selection bias and covers only a subset of the support of input distribution $P_X$ over which we aim to evaluate the model performance. Assuming we have a pool of data $\mathcal{X}_{\mathsf{pool}}$, we design adaptive sampling algorithms that iteratively select inputs in $\mathcal{X}_{\mathsf{pool}}$ to be labeled. Since labeling inputs takes time in practice, we model real-world instances by considering *batched* settings. Our goal is to sequentially label batches of data to accurately estimate model performance over $P_X$ and therefore we assume we have access to a set of inputs $\mathcal{X}_{\mathsf{eval}} \sim P_X$. We use the squared loss to illustrate our framework, where our goal is to evaluate $\mathbb{E}_{X \sim P_X}[(Y - \Psi(X))^2]$. Under the "likelihood" function $p(y|f, x) = p_\varepsilon(y - f(x))$, let $g(f)$ be the performance of the AI model $\Psi(\cdot)$ under the data generating function $f$, which we refer to as our estimand of interest. When we consider the mean squared loss, $g(f)$ is given by

$$g(f) \triangleq \mathbb{E}_{X \sim P_X} \left[ \mathbb{E}_{Y \sim p(\cdot|f,X)} \left[ (Y - \Psi(X))^2 \right] \mid f \right]. \tag{1}$$

Our framework is general and can be extended to other settings. For example, a clinically useful metric is `Recall`, defined as the fraction of individuals that the model $\Psi(\cdot)$ correctly labels as positive among all the individuals who actually have the positive label $g(f) \triangleq \mathbb{E}_{X \sim P_X} \left[ \mathbb{E}_{Y \sim p(\cdot|f,X)} \left[ \mathbf{1}\left\{ \Psi(X) > 0 \right\} | Y = 1 \right] \right]$.

Since the true function $f^\star$ is unknown, we model it from a Bayesian perspective by formulating a posterior given the available supervised data. We refer to uncertainty over the data generating function $f$ as *epistemic* uncertainty—since we can resolve it with more data—and that over the measurement noise $\varepsilon$ as *aleatoric* uncertainty. Assuming independence given features $X$, we model the marginal likelihood of the data via the product $p(\mathcal{Y}|f, \mathcal{X}) = \prod_{(X,Y) \in \mathcal{D}} p(Y|f, X)$ where $\mathcal{D} = (\mathcal{X} \times \mathcal{Y})$. Our prior belief $\mu$ over functions $f$ reflects our uncertainty about how labels are generated given features.

To adaptively label inputs from $\mathcal{X}_{\mathsf{pool}}$, we quantify epistemic uncertainty over the labels reliably using an uncertainty quantification (UQ) method. Roughly speaking, a UQ method provides us the ability to formulate a posterior belief $\mu(f \mid \mathcal{D})$ given any supervised data $\mathcal{D}$. As we detail in Section 4.1, our framework can leverage both classical Bayesian models like Gaussian processes and recent advancements in deep learning-based UQ methods. As new batches are labeled, we update our posterior beliefs about $f$ over time, which we view as "state transitions" of a dynamical system. Recalling the Markov decision process depicted in Figure 2, we sequentially label a batch of inputs from $\mathcal{X}_{\mathsf{pool}}$ (actions), which lead to state transitions (posterior updates). Specifically, our initial state is given by $\mu_0(\cdot) = \mu(\cdot \mid \mathcal{D}^0)$ and at each period $t$, we label a batch of $K_t$ inputs $\mathcal{X}^{t+1} \subset \mathcal{X}_{\mathsf{pool}}$ resulting in labeled data $\mathcal{D}^{t+1} = (\mathcal{X}^{t+1} \times \mathcal{Y}^{t+1})$. After acquiring the labels at each step $t$, we update the posterior state to $\mu_{t+1}(\cdot) = \mu_t(\cdot \mid \mathcal{D}^{t+1})$. Modeling practical instances, we consider a small horizon problem with limited adaptivity $T$. Formulating an MDP over posterior states has long conceptual roots, dating back to the Gittin's index for multi-armed bandits [12].

We denote by $\pi_t$ the adaptive labeling policy at period $t$. We account for randomized policies $\mathcal{X}^{t+1} \sim \pi_t(\mu_t)$ with a flexible batch size $|\mathcal{X}^{t+1}| = K_t$. We assume $\pi_t$ is $\mathcal{F}_t$−measurable for all $t \le T$, where $\mathcal{F}_t$ is the filtration generated by the observations up to the end of step $t$. Observe that $\mu_{t+1}$ contains randomness in the policy $\pi_t$ as well as randomness in $\mathcal{Y}^{t+1} \mid (\mathcal{X}^{t+1}, \mu_t)$. Letting $\pi = \{\pi_0, ...., \pi_{T-1}\}$, we minimize the uncertainty over $g(f)$ at the end of data collection

$$H(\pi) \triangleq \mathbb{E}_{\mathcal{D}^{1:T} \sim \pi} \left[ G(\mu_T) \right] \triangleq \mathbb{E}_{\mathcal{D}^{1:T} \sim \pi} \left[ G(\mu(\cdot \mid \mathcal{D}^{0:T})) \right] = \mathbb{E}_{\mathcal{D}^{1:T} \sim \pi} \left[ \mathrm{Var}_{f \sim \mu(\cdot|\mathcal{D}^{0:T})} g(f) \right], \tag{2}$$

where $G(\mu_T) = \mathrm{Var}_{f \sim \mu_T} g(f)$. In the above objective (2) we assumed that the modeler pays a fixed and equal cost for each outcome. Our framework can seamlessly accommodate variable labeling cost as well. Specifically, we can define a cost function $c(\cdot)$ applied on the selected subsets and update the objective accordingly to have a term $\lambda c(\mathcal{D}^{1:T})$ where $\lambda$ is the penalty factor that controls the trade-off between minimizing variance and cost of acquiring samples.

In the planning problem (2), states are continuous and the action space is combinatorial. To address these issues, we propose a continuous policy parameterization $\pi_\theta$ (with parameter $\theta$) in the next section. We solve this MDP using policy gradients. Let the gradient be defined as $\nabla_\theta H(\theta) \triangleq \nabla_\theta \mathbb{E}_{A \sim \pi_\theta}[G(A)]$. Here, we slightly abuse notations by letting $A \triangleq \mathcal{D}^{1:T}$ and $G(A) \triangleq G(\mu(\cdot \mid \mathcal{D}^0, \mathcal{D}^{1:T}))$, where the distribution of $A$ depends on $\pi_\theta$. To approximately solve this MDP using policy gradients, the score function estimator (REINFORCE [38]) uses the fact

that $\nabla_\theta \mathbb{E}_{A\sim\pi_\theta}[G(A)] = \mathbb{E}_{A\sim\pi_\theta}[G(A)\nabla_\theta \log \pi_\theta(A)]$ for any function $G(\cdot)$. Despite being unbiased, Monte-Carlo estimation of the above expectation typically suffers from prohibitively large variance [33] especially when $\pi_\theta(A)$ is small. Although a stream of work strives to provide variance reduction techniques for REINFORCE [19, 29], they require value function estimates that are also challenging to compute in our planning problem (2).

## 4 Planning using pathwise policy gradients

Our main algorithmic insight is that for systems with known dynamics (posterior updates) or where the dynamics can be approximated sufficiently well, we can leverage auto-differentiation to directly estimate approximate pathwise policy gradients (4) instead of relying on high-variance score-based gradients. The policies derived using our efficient differentiable simulator exploits the system structure and achieves significantly improved performance compared to policies that do not rely on gradient information, as detailed in Section 5. Intuitively, this is akin to finding a random variable $Z \sim p_Z$ distributed independent of policy $\pi_\theta$, such that $A = h(Z,\theta)$ and

$$\nabla_\theta \mathbb{E}_{A\sim\pi_\theta}[G(A)] = \nabla_\theta \mathbb{E}_{Z\sim p_Z}[G(h(Z,\theta))] \stackrel{(a)}{=} \mathbb{E}_{Z\sim p_Z}[\nabla_\theta G(h(Z,\theta))] \tag{3}$$

However, the equality $(a)$ above holds only if $G(h(Z,\theta))$ is differentiable w.r.t. $\theta$. In our case, as we will see later, $h(Z,\theta)$ is non-differentiable w.r.t. $\theta$ because the actions are discrete and combinatorial. To address this, we will use a smoothed approximation, $h_\tau(Z,\theta)$, such that $G(h_\tau(Z,\theta))$ becomes differentiable. Here, $\tau$ is a temperature parameter that controls the degree of smoothing. Consequently, we can approximate the gradient as follows –

$$\nabla_\theta \mathbb{E}_{A\sim\pi_\theta}[G(A)] \approx \nabla_\theta \mathbb{E}[G(h_\tau(Z,\theta))] \approx \frac{1}{n}\sum_{i=1}^n \nabla_\theta G(h_\tau(Z_i,\theta)). \tag{4}$$

It is important to note that sometimes even $G(\cdot)$ is non-differentiable, and in that case we must also consider a smooth approximation of $G(\cdot)$ (e.g., refer to Section D.1 for details on smoothing the `Recall` objective discussed earlier in Section 3). Our approach is inspired by the line of work on differentiable simulators [4, 15, 21, 40, 35]. Theoretical analysis in stochastic optimization [11, 20] also highlights the benefit of these estimators over zeroth-order gradient estimates of a stochastic objective such as REINFORCE. Suh et al. [35] notes that first-order gradient estimators typically perform well when the objective is sufficiently smooth and continuous. In what follows, we empirically demonstrate that through a careful smoothing of the planning objective, it is possible to trade-off bias and variance through auto-differentiation.

---

**Algorithm 1** `Autodiff 1-lookahead`

---

1: **Inputs:** Labeled batch data $\mathcal{D}^0$, horizon $T$, UQ module, pool data $\mathcal{X}_{\text{pool}}$, batch size $K$
  **Returns:** Selected batches $(\mathcal{X}^t, \mathcal{Y}^t)$ for $1 \leq t \leq T$ and updated estimate of the objective $G(\mu_T)$
2: $t = 0$ : Compute initial posterior state $\mu_0$ based on UQ module and batch of labeled data $\mathcal{D}^0$.
3: **for** $0 \leq t \leq T-1$: **do**
4:      $\pi_t \triangleq$ Algorithm 2 (Inputs: Posterior $\mu_t$, pool $(\mathcal{X}_{\text{pool}}^{t+1})$, batch size $K$)
5:      $\mathcal{X}^{t+1} = \{X_j : X_j \in \texttt{SortDescending}(\mathcal{X}_{\text{pool}}^{t+1}; \pi_t) \text{ for } 1 \leq j \leq K\}$
6:      Obtain labels $\mathcal{Y}^{t+1}$ to create $\mathcal{D}^{t+1}$
7:      Update posterior state: $\mu_{t+1} \triangleq \mu_t(\cdot \mid \mathcal{D}^{t+1})$
8:      Estimate the objective $G(\mu_{t+1})$
9: **end for**

---

We showcase the power of *planning using pathwise policy gradients* by considering the simplest possible planning algorithm: one-step lookaheads. Empirically, we demonstrate that even one-step look-ahead achieves significant improvement in sampling efficiency over other heuristic baselines. Additionally, we highlight open engineering directions that can enable efficient multi-step planning methods. At each time step $t$, our algorithm selects a batch of inputs $\mathcal{X}^{t+1}$ to minimize the uncertainty over the one-step target estimand $G(\mu_{t+1})$ based on current posterior belief $\mu_t$. After selecting the batch $\mathcal{X}^{t+1}$, we observe the corresponding labels $\mathcal{Y}^{t+1}$ and update posterior belief over the data-generating function $f$ to $\mu_{t+1}$ (see Algorithm 1). The samples $\mathcal{X}^{t+1}$ to query are determined using a policy $\pi_{t,\theta} \triangleq \pi_\theta$, which we optimize through pathwise policy gradients (Algorithm 2). To optimize

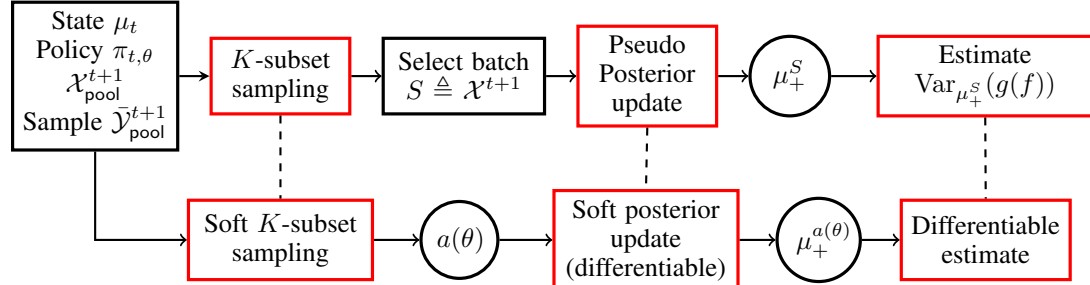

Figure 3: Differentiable one-step look ahead pipeline for efficient adaptive sampling

the policy $\pi_{t,\theta}$, we simulate "roll-outs" of posterior beliefs over $f$ using imagined pseudo labels $\bar{\mathcal{Y}}_{\text{pool}}$ generated from current belief $\mu_t$. Algorithm 1 summarizes each of these steps of the overall procedure. Our conceptual framework is general and can leverage multi-step look-aheads [2, 6, 7].

Our algorithm consists of three major components which we will describe in detail shortly. First, we parameterize a single-batch policy $\pi_\theta$ and use $K$-subset sampling [39] to choose $K$ samples. Second, we use a UQ module that characterizes posterior beliefs over $f$. Finally, to reliably optimize $\theta$, we adopt an auto-differentiable "roll-out" pipeline through which we approximate policy gradients by smoothing the pipeline. The overall pipeline (and its differentiable analogue) is briefly described in Figure 3. We also explain our algorithm graphically in Figure 8 of Section A. Our codebase is available at https://github.com/namkoong-lab/adaptive-labeling.

## 4.1 Uncertainty Quantification (UQ) Module

The UQ module maintains an estimate $\mu_t$ over posterior beliefs of $f$ over time, enabling our planning framework. Our method is agnostic to the UQ module, and we illustrate this using two instantiations: i) Gaussian Processes (GP) that are extensively used in the Bayesian Optimization literature and are known to work well in low dimensional data and regression setting, and ii) recently developed neural network-based UQ methods such as `Ensembles` / `Ensemble+` [27].

**Gaussian Processes** GPs $f(\cdot) \sim \mathcal{GP}(m(\cdot), \mathcal{K}(\cdot, \cdot))$ are defined by a mean function $m(\cdot)$ and kernel $\mathcal{K}(\cdot, \cdot)$, where for any inputs $X$, $f(X)$ is Gaussian with mean $m(X)$ and $\text{Cov}(f(X_i), f(X_j)) = \mathcal{K}(X_i, X_j)$. Additionally, the observation consists of $(X, Y)$ with $Y = f^\star(X) + \varepsilon$ and $\varepsilon \sim \mathcal{N}(0, \sigma^2 I)$ for some noise level $\sigma > 0$. Posterior updates have a closed form and are differentiable, as we review in Section B.

**Ensembles** `Ensembles` [18] learn an ensemble of neural networks from given data, with each network having independently initialized weights. `Ensemble+` [26] extends this approach by combining ensembles of neural networks with randomized prior functions and bootstrap sampling [25]. The prior function is added to each network in the ensemble, trained using L2 regularization. Recently, Epistemic neural networks (ENNs) [28] have also been shown to be an effective way of quantifying uncertainty. All these deep learning models are parametrized by some parameter $\eta$. For a given sample $\{(X_j, Y_j)\}_{j \in \mathcal{I}}$, the model weights $\eta$ are update through gradient descent under a loss function $\ell(.)$, with the update rule expressed as: $\eta_{\text{new}} = \eta - \sum_{j \in \mathcal{I}} \nabla_\eta \ell(X_j, Y_j, \eta)$.

## 4.2 Sampling Policy

At each time step, a batch of $K$ samples is queried to improve the objective. The samples are selected from a pool $\mathcal{X}_{\text{pool}}$ of size $n$, based on weights $\boldsymbol{w}(\theta)$. Specifically, given weights $\boldsymbol{w} \geq \boldsymbol{0}$ and a batch size $K$, the $K$-subset sampling mechanism generates a random vector $S \in \{0, 1\}^n$, whose distribution depends on $\boldsymbol{w}$ such that $\sum_{i=1}^n S_i = K$. Let $\boldsymbol{e}^j$ denote the $j$-th unit vector, and consider a sequence of unit vectors $[\boldsymbol{e}^{i_1} \cdots \boldsymbol{e}^{i_k}] \triangleq s$. Using a parametrization known as weighted reservoir sampling (wrs) [39], the probability of selecting a sequence $s$ is given by:

$$p_{\text{wrs}}([\boldsymbol{e}^{i_1}, \cdots, \boldsymbol{e}^{i_k}] | \boldsymbol{w}) \triangleq \frac{w_{i_1}}{\sum_{j=1}^n w_j} \frac{w_{i_2}}{\sum_{j=1}^n w_j - w_{i_1}} \cdots \frac{w_{i_k}}{\sum_{j=1}^n w_j - \sum_{j=1}^{k-1} w_{i_j}}. \tag{5}$$

The probability of selecting a batch $S$ can then be defined as: $p(S | \boldsymbol{w}) \triangleq \sum_{s \in \Pi(S)} p_{\text{wrs}}(s | \boldsymbol{w})$, where $\Pi(S) = \left\{ s : S = \sum_{j=1}^k s[j] \right\}$ denote all sequences of unit vectors with sum equal to $S$.

---

**Algorithm 2** One-step look-ahead policy gradient

---

1: **Inputs:** Posterior belief $\mu$, pool data $(\mathcal{X}_{\mathsf{pool}})$, batch size $K$, training epochs: $M$
   **Returns:** Policy $\pi_\theta$
2: Initialize the policy $\pi_\theta$ (parametrized by $\theta$)
3: **for** $1 \leq m \leq M$: **do**
4:    Generate peusdo labels $\bar{\mathcal{Y}}_{\mathsf{pool}}$ for all the $\mathcal{X}_{\mathsf{pool}}$, where $\bar{\mathcal{Y}}_{\mathsf{pool}}$ is drawn from the current posterior state $\mu$.
5:    Use Algorithm 3 (Input: $\pi_\theta := \boldsymbol{w}(\theta)$) to generate a random soft vector $\boldsymbol{a}(\theta)$ corresponding to policy $\pi_\theta$
6:    Update (pseudo) the posterior state to $\mu_+^{\boldsymbol{a}(\theta)}$ using the pool data and pseudo labels $(\mathcal{X}_{\mathsf{pool}}, \bar{\mathcal{Y}}_{\mathsf{pool}})$
7: Estimate the objective $G(\mu_+^{\boldsymbol{a}(\theta)})$
8: SGD update: $\theta \leftarrow \theta - \epsilon_t \nabla_\theta G(\mu_+^{\boldsymbol{a}(\theta)})$
9: **end for**
10: **Return:** Policy $\pi_\theta$

---

## 4.3 Differentiable pipeline

We propose a fully differentiable pipeline by ensuring that each component of the pipeline: i) $K$-subset sampling, ii) posterior updates (UQ module), and iii) the objective are differentiable. We use parametrizations $\theta$ for the policy function $\pi$ which yields sampling probabilities $\boldsymbol{w}(\theta)$. Posterior beliefs $\mu$ are parametrized by $\eta$ (e.g., `Ensemble+` weights). In the following, we describe how we smooth the respective components. The one-step objective parametrized by $\theta$ is given by

$$H(\theta) \triangleq \mathbb{E}_{\bar{\mathcal{Y}}_{\mathsf{pool}} \sim \mu; S \sim p(\cdot|\boldsymbol{w}(\theta))}[G(\mu_+^S)] \tag{6}$$

where $G$ was defined in (2). Here, $p(\cdot|\boldsymbol{w}(\theta))$ is the distribution over subsets governed by $\boldsymbol{w}$ as defined by (5). Further, $\mu$ is the current posterior state and $\mu_+$ is the updated posterior state after incorporating the the additional batch $S$ selected from $\mathcal{X}_{\mathsf{pool}}$ and the corresponding pseudo-outcomes from $\bar{\mathcal{Y}}_{\mathsf{pool}}$ (drawn based on current posterior state $\mu$). We smooth the objective $H(\theta)$ to estimate $\nabla_\theta H(\theta)$ using a soft $K$-subset sampling.

**Soft $K$-subset sampling** We use the smoothed $K$-subset sampling procedure proposed in Xie and Ermon [39]. Briefly, the Algorithm 3 introduced in [39] produces a random vector $\boldsymbol{a}$ such that $\sum_{i=1}^n a_i = K$ (soft version of $S$), such that $\boldsymbol{a}(\theta) \sim p(\cdot|\boldsymbol{w}(\theta)) \equiv \pi_\theta(\cdot)$ (5), and is differentiable. We defer the details to Section C.

**Smoothing the UQ module** To enable differentiable posterior updates, we approximate the posterior updates using the soft $K$-subset samples $\boldsymbol{a}(\theta)$. In the case of GPs, the soft $K$-subset selection variable $a$ can be interpreted as a weighting of samples in a GP. Thus, closed form GP updates can be analogously used with appropriate weighting, described in the Appendix D. To perform weighted updates for deep learning based UQ modules (`Ensembles` / `Ensemble+` ), we approximate previous gradient updates by incorporating soft sample weights $\boldsymbol{a}(\theta)$:

$$\eta_1(\boldsymbol{a}) \triangleq \eta_0 - \sum_j a_j(\theta) \nabla_\eta \ell(X_j, \bar{Y}_j, \eta), \tag{7}$$

where $\ell$ is the loss function of the concerned model, with gradients evaluated at $\eta$. We interpret $\eta_1$ as an approximation of the optimal UQ module parameter with one gradient step. Similarly, we define a higher-order approximation $\eta_{h+1}(\boldsymbol{a})$ as

$$\eta_{h+1}(\boldsymbol{a}) \triangleq \eta_h(\boldsymbol{a}(\theta)) - \sum_j a_j(\theta) \nabla_\eta \ell(X_j, \bar{Y}_j, \eta) \mid_{\eta=\eta_h(\boldsymbol{a}(\theta))} . \tag{8}$$

Note here that effectively, the gradient is now of the form $\nabla_\theta G\left(\operatorname{argmin}_\eta \ell(\eta(\boldsymbol{a}(\theta)))\right) \equiv \nabla_\theta G(\eta_+(\boldsymbol{a}(\theta)))$. To compute this, we use higher [13] and torchopt [32] which allows us to approximately differentiate through the argmin. In practice, there is a trade-off between the computational time and the accuracy of the gradient approximation depending on the number of training steps we do to estimate the argmin.

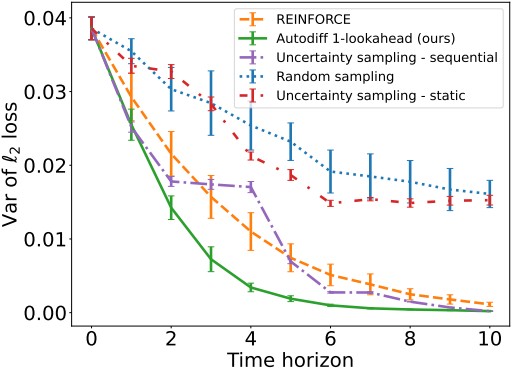 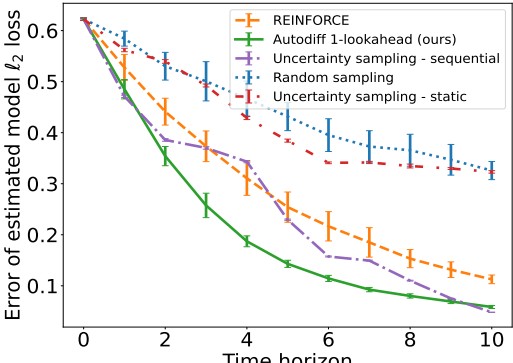

Figure 4: (Synthetic data) Variance of mean squared loss evaluated through the posterior belief $\mu_t$ at each horizon $t$. This is the objective that policy gradient methods like REINFORCE and `Autodiff 1-lookahead` optimizes. 1-step lookaheads are surprisingly effective even in long horizons.

Figure 5: (Synthetic data) Error between MSE calculated based on collected data $\mathcal{D}^{0:T}$ vs. population oracle MSE over $\mathcal{D}_{\text{eval}} \sim P_X$. Reducing uncertainty over posteriors directly leads to better OOD evaluations. 1-step lookaheads significantly outperform active learning heuristics in small horizons.

## 5 Experiments

We empirically demonstrate effectiveness of our planning framework on both synthetic and real datasets by focusing on the simplest planning algorithm: 1-step lookaheads. Using two uncertainty quantification modules—GPs and `Ensembles` / `Ensemble+` —we show that planning with pathwise gradients is a promising approach to adaptive labeling. Throughout this section, we focus on evaluating the mean squared error of a regression model $\Psi$, and develop adaptive policies that minimize uncertainty on this quantity ($g(f)$). When GPs provide a valid model of uncertainty, our experiments show that our auto-differentiable planning framework significantly outperforms other baselines. We further demonstrate that our conceptual framework extends to deep learning-based uncertainty quantification methods like `Ensemble+` while highlighting computational challenges that need to be resolved in order to scale our ideas. For simplicity, we assume a naive predictor, i.e., $\psi(\cdot) \equiv 0$. However, we emphasize that this problem is just as complex as if we were using a sophisticated model $\psi(.)$. The performance gap between the algorithms primarily depends on the level of uncertainty in our prior beliefs.

To evaluate the performance of our algorithm, we benchmark it against several baselines. Our first set of baselines are from active learning [1]:

**Active Learning Heuristics: (1)** UNCERTAINTY SAMPLING (STATIC): In this approach, we query the samples for which the model is least certain about. Specifically, we estimate the variance of the latent output $f(X)$ for each $X \in \mathcal{X}_{\text{pool}}$ using the UQ module and select the top-$K$ points with the highest uncertainty. **(2)** UNCERTAINTY SAMPLING (SEQUENTIAL): This is a greedy heuristic that sequentially selects the points with the highest uncertainty within a batch, while updating the posterior beliefs using pseudo labels from the current posterior state. Unlike UNCERTAINTY SAMPLING (STATIC), this method takes into account the information gained from each point within batch, and hence tries to diversify the selected points within a batch.

We also compare our approach to solving the planning problem using **(3)** REINFORCE-based policy gradients, which implements one-step look ahead policy gradient using the score trick. Finally, we study **(4)** RANDOM SAMPLING, which selects each batch uniformly at random from the pool. We repeat all experiments with 10 random seeds.

### 5.1 Planning with Gaussian processes

We now briefly describe the data generation process for the GP experiments, deferring a more detailed discussion of the dataset generation to Section E. We use both the synthetic data and the real data to test our methodology. For the *simulated data*, we construct a setting where the general population is distributed across *51 non-overlapping clusters* while the initial labeled data $\mathcal{D}^0$ just comes from one cluster. In contrast, both $\mathcal{D}_{\text{pool}} \triangleq (\mathcal{X}_{\text{pool}}, Y_{\text{pool}})$, $\mathcal{D}_{\text{eval}} \triangleq (\mathcal{X}_{\text{eval}}, \mathcal{Y}_{\text{eval}})$ are generated from all the clusters. We begin with a low-dimensional scenario, generating a one-dimensional regression setting using a Gaussian Process (GP). Although the data-generating process is not known to the

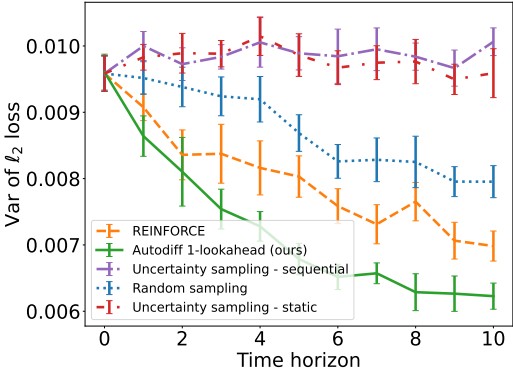
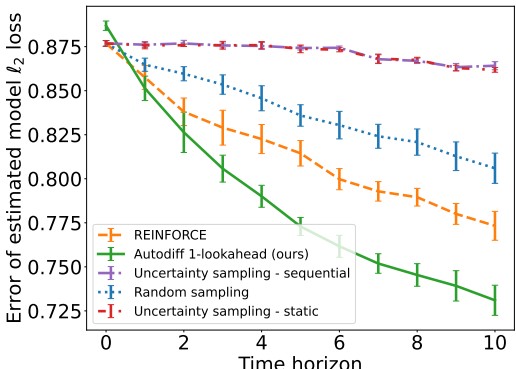

Figure 6: (Real-world eICU data) Variance of mean squared loss evaluated through the posterior belief $\mu_t$ at each horizon $t$. Even 1-step lookaheads are extremely effective planners, and auto-differentiation-based pathwise policy gradients provide a reliable optimization algorithm based on low-variance gradients.

Figure 7: (Real-world eICU data) Error between MSE calculated based on collected data $\mathcal{D}^{0:T}$ vs. population oracle MSE over $\mathcal{D}_{\text{eval}} \sim P_X$. Reducing uncertainty over posteriors directly leads to better OOD evaluations. Our method significantly outperforms active learning heuristics, and 1-lookahead with REINFORCE policy gradients.

algorithms, we assume that the GP hyperparameters are known to all the algorithms to ensure fair comparisons. This can be viewed as a setting where our prior is well-specified, allowing us to isolate the effects of different policy optimization approaches without any concerns about the misspecified priors or in-accurate posteriors. We select 10 batches, each of size $K = 5$ across $T = 10$ time horizons.

To examine the robustness of our method against the distributional assumptions made in the simulated case, we then move to a real dataset where the correct prior is not known. We simulate selection bias from the eICU dataset [30], which contains real-world patient data with in-hospital mortality outcomes. We conduct a $k$-means clustering to generate 51 clusters and then select data from those clusters. We view this to be a credible replication of practice, as severe distribution shifts are common due to selection bias in clinical labels. To convert the binary mortality labels into a regression setting, we train a random forest classifier and fit a GP on predicted scores, which serves as the UQ module for all the algorithms. As before, the task is to select 10 batches, each consisting of 5 samples, across 10 time horizons.

In Figures 4 and 5, we present results for the simulated data. Figure 4 shows the variance of $\ell_2$ loss, while Figure 5 presents the error in the estimated $\ell_2$ loss using $\mu_t$ (relative to true $\ell_2$ loss, that is unknown to the algorithm). As we can see from these plots, our method `Autodiff 1-lookahead` gives substantial improvements over active learning baselines. Compared to the other one-step lookahead planning approach using REINFORCE-based policy gradients, we observe that pathwise policy gradients provide significantly more robust performance over all horizons.

In Figures 6 and 7, we observe similar findings on the eICU data. We see that planning policies (REINFORCE and ours) consistently outperform other heuristics by a large margin. Active learning baselines perform poorly in these small-horizon batched problems and can sometimes be even worse than the random search baselines. Overall, our results show the importance of careful planning in adaptive labeling for reliable model evaluation.

We offer some intuition as to why `Autodiff 1-lookahead` may outperform other heuristic algorithms. First, UNCERTAINTY SAMPLING (STATIC) while myopically selects the top-$K$ inputs with the highest uncertainty, it fails to consider the overlap in information content among the "best" instances; see [1] for more details. In other words, it might acquire points from the same region with high uncertainty while failing to induce diversity among the batch. Although UNCERTAINTY SAMPLING (SEQUENTIAL) somewhat addresses the issue of information overlap, a significant drawback of this algorithm is the disconnect between the objective we aim to optimize and the algorithm. For example, it might sample from a region with high uncertainty but very low density.

### 5.2 Planning with neural network-based uncertainty quantification methods (`Ensemble+`)

We now provide a proof-of-concept that shows the generalizability of our conceptual framework to the deep learning based UQ modules, specifically focusing on `Ensemble+` due to their previously

observed superior performance [28]. Recall that implementing our framework with deep learning based UQ modules requires us to retrain the model across multiple possible random actions $\boldsymbol{a}(\theta)$ sampled from the current policy $\pi_\theta$. This requires significant computational resources, in sharp contrast to the GPs where the posteriors are in closed form and can be readily updated and differentiated.

Due to the computational constraints, we test `Ensemble+` on a toy setting to demonstrate the generalizability of our framework. We consider a setting where the general population consists of four clusters, while the initial labeled data only comes from one cluster. Again we generate data using GPs. The task is to select a batch of 2 points in one horizon. We detail the `Ensemble+` architecture in the Appendix, and we assume prior uncertainty to be large (depend on the scaling of prior generating functions). The results are summarized in the Table 1.

Table 1: Performance under `Ensemble+` as UQ module

| Algorithm | Variance of $\ell_2$ loss estimate | Error of $\ell_2$ loss estimate |
|---|---|---|
| RANDOM SAMPLING | $7129.8 \pm 1027.0$ | $136.2 \pm 8.28$ |
| UNCERTAINTY SAMPLING (STATIC) | $10852 \pm 0.0$ | $162.156 \pm 0.0$ |
| UNCERTAINTY SAMPLING (SEQUENTIAL) | $8585.5 \pm 898.9$ | $144 \pm 6.93$ |
| REINFORCE | $1697.1 \pm 0.0$ | $45.27 \pm 0.0$ |
| `Autodiff 1-lookahead` | $1697.1 \pm 0.0$ | $45.27 \pm 0.0$ |

## 6 Gradient Estimation - Theoretical Insight

In this section, we present theoretical analysis of the statistical properties (bias-variance trade-off) of the REINFORCE and AUTODIFF (smoothed-pathwise (4)) gradient estimators. To compare the two estimators, we consider a simplified setting with two actions $\mathcal{A} = \{-1, 1\}$, and the policy is parametrized by $\theta \in [0, 1]$ with $\pi_\theta(-1) = \theta$ and $\pi_\theta(1) = 1 - \theta$, Additionally, we are interested in the gradient $\nabla_\theta H(\theta)$, where $H(\theta) = \mathbb{E}_{A \sim \pi_\theta} G(A)$ with $G(A) = A$ for simplicity. Given $N$ i.i.d. samples of $A_i \sim \pi_\theta$, the REINFORCE estimator is defined as $\hat{\nabla}_N^{\mathsf{RF}} = \frac{1}{N} \sum_{i=1}^N \left( G(A_i) \nabla_\theta \log(\pi_\theta(A_i)) \right)$. The pathwise gradient estimator $\nabla_{\tau,N}^{\mathsf{grad}}$ is the $N$-sample approximation of $\mathbb{E}_{U \sim \mathsf{Uni}[0,1]}[\nabla_\theta G(h_\tau(U, \theta))]$, where $h_\tau(U, \theta) = \left( \exp\left(\frac{U-\theta}{\tau}\right) - 1 \right) / \left( \exp\left(\frac{U-\theta}{\tau}\right) + 1 \right)$ (smoothing of $h(U, \theta) := 2\mathcal{I}(U > \theta) - 1 \overset{d}{=} A$). Our result (proof in Section F) highlights the conditions under which $\hat{\nabla}_{\tau,N}^{\mathsf{grad}}$ achieves a lower mean squared error (mse) compared to $\hat{\nabla}_N^{\mathsf{RF}}$.

**Theorem 1.** *For $\theta \in [0, 1]$ and $N \leq \frac{1}{4\theta(1-\theta)} - 1$, there exists $\tilde{\tau}$ depending on $(N, \theta)$ such that*

$$\mathsf{MSE}(\hat{\nabla}_{\tilde{\tau},N}^{\mathsf{grad}}) < 4 \leq \mathsf{MSE}(\hat{\nabla}_N^{\mathsf{RF}}).$$

*Additionally, for any $N$, $\theta = \frac{1}{k}$, and $k \to \infty$, we have the following $\mathsf{MSE}(\hat{\nabla}_N^{\mathsf{RF}}) = \Omega(k)$, $\mathsf{MSE}(\hat{\nabla}_{\tilde{\tau},N}^{\mathsf{grad}}) < 4$. The same statement holds for $\theta = 1 - \frac{1}{k}$ as well. This implies that the mse of $\hat{\nabla}_N^{\mathsf{RF}}$ is unbounded, while the mse of gradient estimator is bounded.*

## 7 Conclusion and Future Work

Supervised data often suffers severe selection bias when labels are expensive. We propose a new framework for adaptive labeling for model evaluation under out-of-support distribution shifts. Following a Bayesian framework, we formulate an MDP over posterior beliefs on model performance. We show that these planning problems can be efficiently solved with pathwise policy gradients, computed through a carefully designed auto-differentiable pipeline. We also demonstrate that even one-step lookahead policies outperform heuristic active learning algorithms. However, this improvement comes at the cost of additional computational resources. Our formulation is thus appropriate in high-stake settings such as clinical trials where labeling costs far exceed computational costs.

We further highlight some important nuances which we learned from our experiments. There are some important properties that a UQ module should have for it to perform well in our framework. The most important feature is that posteriors should be consistent, that is, early stopping of the training should not cause a significant change across the ranking of the subsets of the pool. Second, the posterior update should be done efficiently, which can be achieved either by doing parallelization or having UQ modules that provide readily available posterior updates, such as GPs. Recent advances in Bayesian transformers [24, 22] will be an interesting direction to explore in this regard.

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

# A    Graphical Representation of Our Algorithm

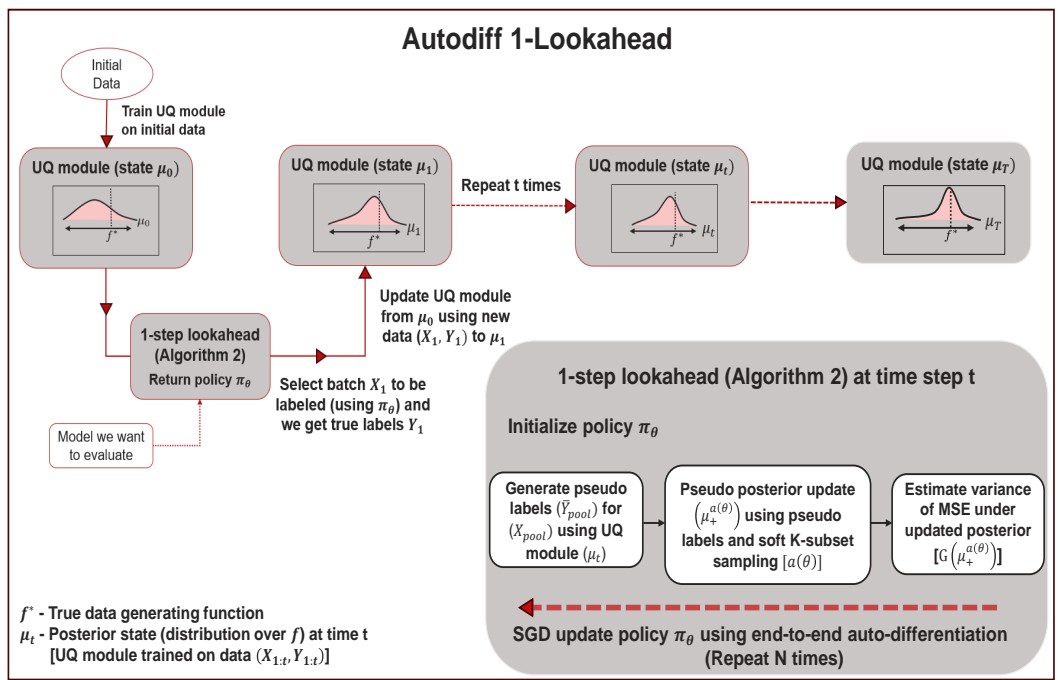

Figure 8: Description of our algorithm – `Autodiff 1-lookahead`

# B    Gaussian Process posterior updates

For any inputs $\boldsymbol{X}$, we assume $f(\boldsymbol{X})$ is Gaussian with mean $m(\boldsymbol{X})$ and $\mathrm{Cov}(f(X_i), f(X_j)) = \mathcal{K}(X_i, X_j)$. In addition, the observation consists of $(\boldsymbol{X}, \boldsymbol{Y})$ with $\boldsymbol{Y} = f^\star(\boldsymbol{X}) + \epsilon$ and $\epsilon \sim \mathcal{N}(0, \sigma^2 I)$ for some noise level $\sigma > 0$. Given the training data $(\boldsymbol{X}, \boldsymbol{Y})$ and test points $X^*$, closed form posterior estimates over $f^\star$ can be computed

$$\boldsymbol{f}^* \mid \boldsymbol{X}, \boldsymbol{Y}, \boldsymbol{X^*} \sim \mathcal{N}(\bar{\boldsymbol{f}}|^*, \mathrm{Cov}(\boldsymbol{f}^*)),$$
$$\text{where } \bar{\boldsymbol{f}}|^* \triangleq \mathcal{K}(\boldsymbol{X^*}, \boldsymbol{X})[\mathcal{K}(\boldsymbol{X}, \boldsymbol{X}) + \sigma^2 I]^{-1} Y$$
$$\mathrm{Cov}(\boldsymbol{f}^*) \triangleq \mathcal{K}(\boldsymbol{X^*}, \boldsymbol{X^*}) - \mathcal{K}(\boldsymbol{X^*}, \boldsymbol{X})[\mathcal{K}(\boldsymbol{X}, \boldsymbol{X}) + \sigma^2 I]^{-1} \mathcal{K}(\boldsymbol{X}, \boldsymbol{X^*}).$$

Using the above expression, we can update our belief about $\boldsymbol{f}^*$ after observing new data.

# C    Details of $K$-subset sampling algorithm

Algorithm 3 for soft $K$-subset sampling was introduced in [39]. We present this algorithm here and refer the reader to [39] for further details.

# D    Details of weighted GP

We present a weighted GP algorithm (Algorithm 4) adapted from Algorithm 2.1 in [31]. Recall that GPs $f(\cdot) \sim \mathcal{GP}(m(\cdot), \mathcal{K}(\cdot, \cdot))$ are defined by a mean function $m(\cdot)$ and kernel $\mathcal{K}(\cdot, \cdot)$.

Now assume we are given input data $(\boldsymbol{X}, \boldsymbol{Y})$, we are also given some weights $\boldsymbol{w}$ for these inputs, and we aim to compute posterior estimates over $\boldsymbol{f}^\star$ for test points $\boldsymbol{X}^*$ considering the weights $\boldsymbol{w}$. Let us define the following terms: $K \triangleq \mathcal{K}(\boldsymbol{X}, \boldsymbol{X}), K_* \triangleq \mathcal{K}(\boldsymbol{X^*}, \boldsymbol{X})$ and $K_{*,*} \triangleq \mathcal{K}(\boldsymbol{X^*}, \boldsymbol{X^*})$. Using these definitions, GP Algorithm 4 provides posterior estimates of $\bar{\boldsymbol{f}}^\star$ for the test samples $\boldsymbol{X}^*$, given that the input data $(\boldsymbol{X}, \boldsymbol{Y})$ has weights $\boldsymbol{w}$.

---

**Algorithm 3** Soft $K$-subset sampling algorithm

---

1: **Inputs:** Weight vector $\boldsymbol{w} \in \mathbb{R}_+^n$
2: Sample $n$ independent standard Gumbel random variables $g_1, g_2, \ldots, g_n$
3: Compute keys $\hat{r}_i = g_i + \log(w_i)$ for all $i$
4: Initialize $\kappa_i^1 = \hat{r}_i$ for all $i = 1, \ldots, n$
5: For $j = 1, .., K$, set

$$a_i^j = \frac{\exp(\kappa_i^j/\tau)}{\sum_{k=1}^n \exp(\kappa_k^j/\tau)} \text{ for all } i = 1, \ldots, n,$$

   $\kappa_i^{j+1} = \kappa_i^j + \log(1 - a_i^j)$ for all $i = 1, ..., n$.
6: **Return:** The soft $K$-hot vector, $\boldsymbol{a} = \boldsymbol{a}^1 + \boldsymbol{a}^2 + ... + \boldsymbol{a}^K$.

---

---

**Algorithm 4** Weighted Gaussian process regression

---

1: **Inputs:** $K, K_*, K_{*,*}, \boldsymbol{Y}, \boldsymbol{w}$
2: $K_{\boldsymbol{w}} = K((\mathbf{1} - I)\boldsymbol{w}\boldsymbol{w}^\top + I)$
3: $K_{\boldsymbol{w},*} = \boldsymbol{w} \odot K_*$
4: $L = \mathsf{Cholesky}(K_{\boldsymbol{w}} + \sigma^2 I)$
5: $\alpha = L^\top \backslash (L \backslash \boldsymbol{Y})$
6: $\bar{f} = K_{\boldsymbol{w},*}^\top \alpha$
7: $v = L \backslash K_{\boldsymbol{w},*}$
8: $V = K_{*,*} - v^\top v$
9: **Return:** mean and covariance: $(\bar{f}, V)$

---

### D.1  Smoothing the recall objective

Performance metric, such as the Recall of a model $\Psi(.)$, is not differentiable. Recall that $H(\theta) \triangleq \mathbb{E}_{\bar{\mathcal{Y}}_{\text{pool}} \sim \mu; S \sim p(\cdot|\boldsymbol{w}(\theta))}[G(\mu_+^S)]$, where $G(\mu_+^S) = \text{Var}_{f \sim \mu_+^S} g(f)$, and $g(f) \triangleq \mathbb{E}_{X \sim P_X}\left[\mathbb{E}_{Y \sim p(\cdot|f,X)}\left[\mathbf{1}\{\Psi(X) > 0\} | Y = 1\right]\right]$ with $\mu_T$ depending on $\pi_\theta$. To estimate $\nabla_\theta H(\theta)$ in a differentiable manner, we can use a smooth approximation of $g(f)$ using the softmax trick [16] and draw $2n$ i.i.d. Gumbel$(0, 1)$ samples $R_i^l$ with $l \in \{1, 2\}$. Define:

$$g_\tau(f; \boldsymbol{R}) \triangleq \frac{\sum_{i=1}^n Y_i^\tau(f; \boldsymbol{R})\Psi(X_i)}{\sum_{i=1}^n Y_i^\tau(f; \boldsymbol{R})},$$

$$\text{where } Y_i^\tau(f; \boldsymbol{R}) = \frac{\exp\left((\log f_i) + R_i^1)/\tau\right)}{\exp\left((\log f_i) + R_i^1)/\tau\right) + \exp\left((\log(1 - f_i)) + R_i^2)/\tau\right)},$$

here $\tau > 0$ is the temperature hyperparameter that controls the smoothing of $g_\tau(f; \boldsymbol{R})$. It is easy to show that $\mathbb{E}_{\boldsymbol{R}}[Y_i^\tau(f; \boldsymbol{R})] \approx f_i$, making $g_\tau(f; \boldsymbol{R})$ a natural approximation of the previously defined Recall $g(f)$.

## E  Experimental details

In this section, we provide detailed information about the experiments discussed in Section 5.

### E.1  Planning with Gaussian Processes - synthetic data experiments

As mentioned earlier, we generate our data using a Gaussian Process (GP). Specifically, we use a GP with an RBF kernel: $f_i \sim \mathcal{GP}(m, \mathcal{K})$, where $m(X) = 0$ and $\mathcal{K}(X, X') = \sigma_f^2 \exp\left(-\frac{||X - X'||_2^2}{2\ell^2}\right)$, further Gaussian noise $N(0, \sigma^2)$ is added to the outputs. We set $\ell = 1$, $\sigma_f^2 = 0.69$, and $\sigma^2 = 0.01$. Recall that the marginal distribution $P_X$ consists of 51 *non-overlapping* clusters. To achieve this, we create a polyadic sampler, which first samples 51 anchor points spaced at linearly increasing distances from the center to avoid overlap. These anchor points serve as our cluster centers. We

sample the points around these anchor points by adding a small Gaussian noise $N(0, 0.25)$. The training points are drawn from a single cluster, while the pool and evaluation points are drawn from all the 51 clusters. The setup includes 100 initial labeled data points, 500 pool points and 285 evaluation points used to estimate the objective.

`Autodiff 1-lookahead` **training and hyperparameters:** As mentioned earlier, the underlying uncertainty quantification (UQ) module for these experiments is the GP. Although our algorithm does not know the true data generating function, but it has access to the GP hyperparameters. Therefore we use a $\mathcal{GP}$ with an RBF kernel: $f_i \sim \mathcal{GP}(m, \mathcal{K})$, where $m(X) = 0$ and $\mathcal{K}(X, X') = \sigma_f^2 \exp\left(-\frac{||X-X'||_2^2}{2\ell^2}\right)$, with Gaussian noise $N(0, \sigma^2)$ added to the outputs. We set $\ell = 1$, $\sigma_f^2 = 0.69$ and $\sigma^2 = 0.01$. For soft $K$-subset sampling (see Algorithm 3), we set $\tau$ to 0.1. Further for evaluating the objective function $\text{Var}(g(f))$ - we take 100 samples of $f(\mathcal{X}_{\text{eval}})$ from the posterior state $\mu_+^{a(\theta)}$, see Algorithm 2. For policy optimization in each horizon, we use the Adam optimizer with a learning rate of 0.1 to perform policy gradient steps over 100 epochs. The results presented are averaged over 10 different training seeds.

**Policy gradient (REINFORCE) training and hyperparameters:** As mentioned earlier, all the algorithms have access to the true to GP hyperparameters. For evaluating the objective $\text{Var}(g(f))$ under a given action, we take 100 samples of $f(\mathcal{X}_{\text{eval}})$. To optimize the policy in each horizon, we use the Adam optimizer with a learning rate of 0.1 to perform policy gradient updates over 100 epochs. The results presented are averaged over 10 different training seeds.

### E.2 Planning with Gaussian Processes - real data (eICU) experiments

The eICU dataset is a healthcare dataset that contains data from various critical care units across the United States. To create a supervised learning setup from this dataset, we first extracted the 10 most important features using a `Random Forest` classifier (number of trees $= 100$, criterion = "gini"). The outcome variable was in-hospital mortality. Some examples of the extracted features include: Hospital length of stay, Number of days on the ventilator and the Last recorded temperature on Day 1 of ICU admission. We transformed the classification outcome variable (in-hospital mortality) into a regression task using the probability of in-hospital mortality predicted by the classifier. To adapt the extracted data to our setting, we introduced selection bias in the data. We generated 51 clusters through the standard $k-$means algorithm. Our initial labeled data comes from only 1 cluster, while the pool and evaluation data comes from all the 51 clusters. Our dataset consists 100 initial labeled data points, 500 pool points, and 285 evaluation points used to estimate the objective.

We then fitted a GP with an RBF kernel to above data using GP-regression. The resulting GP hyperparameters were: $m(x) = 0.255$, $\ell = 0.50$, $\sigma_f^2 = 1.0$, $\sigma^2 = 0.0001$. This $\mathcal{GP}$ model serves as our uncertainty quantification (UQ) module.

`Autodiff 1-lookahead` **training and hyperparameters:** For soft $K$-subset sampling (see Algorithm 3), we set $\tau$ to 0.1. Further to evaluate the objective function $\text{Var}(g(f))$, we take 100 samples of $f(\mathcal{X}_{\text{eval}})$ from the posterior state $\mu_+^{a(\theta)}$, see Algorithm 2. For policy optimization in each horizon, we use the Adam optimizer with a learning rate of 0.1 to perform policy gradient steps over 1000 epochs. The results presented are averaged over 10 different training seeds.

**Policy gradient (REINFORCE) training and hyperparameters:** To evaluate the objective $\text{Var}(g(f))$ under an action, we take 100 samples of $f(\mathcal{X}_{\text{eval}})$. For optimizing the policy in each horizon, we use an Adam optimizer with learning rate of 0.1, performing policy gradient updates over 1000 epochs. The results presented are averaged over 10 different training seeds.

### E.3 Planning with `Ensemble+` experiments

Once again, we use GPs as our data generating process. Specifically, we use a GP with an RBF kernel: $f_i \sim \mathcal{GP}(m, \mathcal{K})$, where $m(X) = 0$ and $\mathcal{K}(X, X') = \sigma_f^2 \exp\left(-\frac{||X-X'||_2^2}{2\ell^2}\right)$. Additionally, Gaussian noise $N(0, \sigma^2)$ is added to the outputs. We set $\ell = 1$, $\sigma_f^2 = 0.69$, and $\sigma^2 = 0.01$. The marginal distribution $P_X$ consists of 4 *non-overlapping* clusters. We follow the same process as in the Gaussian process experiments to form the cluster centers and sample points around them. While the training points are drawn from a single cluster, both the pool and evaluation points are drawn

from all the $4$ clusters. Our setup includes $20$ initial labeled data points, $10$ pool points, and $252$ evaluation points used to evaluate the objective.

`Ensemble+` **Architecture:** We use an ensemble of 10 models. Each model being a 2-hidden layer MLP with 50 units per hidden layer. In addition, each models includes an additive prior. The additive prior function for each model is fixed to be a 2-hidden layer MLP with 50 units in each hidden layer. Specifically, the $m$-th model for $1 \leq m \leq 10$ of the `Ensemble+` takes the form

$$f_{\eta_m}(X) = g_{\eta_m}(X) + \alpha p_m(X),$$

where $g_{\eta_m}$ is the trainable part of the network while $\alpha p_m(\cdot)$ is the additive prior function. Here $\alpha$ controls our prior belief about the uncertainty —higher the $\alpha$, the greater the uncertainty. The prior functions $p_m(\cdot)$ differs across models $m$ due to different initializations. In our setup, we set $\alpha = 100$ to reflect a high level of uncertainty in the prior beliefs.

`Ensemble+` **training:** For a given dataset $\mathcal{D} = (X_i, Y_i)_{i=1}^n$, we train the $m$-th model so as to minimize the following loss function

$$\ell(\eta_m, \mathcal{D}) = \frac{1}{n} \sum_{i=1}^n (g_{\eta_m}(X_i) + \alpha p_m(X_i) - Y_i)^2 + \lambda \|\eta_m\|_2^2.$$

We tune $\lambda$ to $0.1$ and use the Adam optimizer with a tuned learning rate of $0.1$. Each model is trained for 50 iterations. Although in standard `Ensemble+` each model is trained on a different bootstrapped subset from the dataset $\mathcal{D}$, in our case we train all the models within `Ensemble+` on the entire dataset available.

`Autodiff 1-lookahead` **training and hyperparameters:** Recall that in soft $K$-subset sampling there is a hyperparameter $\tau$ (see Algorithm 3), which we set to $0.1$. To differentiate through the argmin operation, we employ the differentiable optimizer from the torchopt package, specifically the MetAdam Optimizer with a learning rate of $0.1$. For optimizing the sampling policy, we use the Adam optimizer with a learning rate of $0.05$, performing policy gradient updates over $500$ epochs.

## F   Proof of Theorem 1

We begin by analyzing the REINFORCE based gradient estimator. With $N$ samples, we have:

$$\mathsf{MSE}(\hat{\nabla}_N^{\mathsf{RF}}) = \mathsf{Var}(\hat{\nabla}_N^{\mathsf{RF}}) = \frac{1}{N} \mathsf{Var}_{A \sim \pi_\theta}\left(A \nabla_\theta \log(\pi_\theta(A))\right).$$

Using the definition of $A$, we derive the following:

$$\mathsf{Var}_{A \sim \pi_\theta}\left(A \nabla_\theta \log(\pi_\theta(A))\right) = \frac{1}{(1-\theta)\theta} - 4.$$

As a result, we have:

$$\mathsf{MSE}(\hat{\nabla}_N^{\mathsf{RF}}) = \frac{1}{N}\left(\frac{1}{(1-\theta)\theta} - 4\right). \tag{9}$$

Next, we analyze the pathwise gradient estimator $\hat{\nabla}_{\tilde{\tau},N}^{\mathsf{grad}}$. Let $h_\tau(U, \theta) = \frac{\exp\left(\frac{U-\theta}{\tau}\right)-1}{\exp\left(\frac{U-\theta}{\tau}\right)+1}$ be a random variable, where $U \sim \mathsf{Uni}[0,1]$, so that:

$$\mathsf{MSE}(\hat{\nabla}_{\tau,N}^{\mathsf{grad}}) = \left[\mathsf{Bias}(\nabla_\theta h_\tau(U, \theta))\right]^2 + \frac{1}{N} \mathsf{Var}(\nabla_\theta h_\tau(U, \theta)). \tag{10}$$

To compute the bias term in (10), we note that

$$\mathbb{E}(h_\tau(U, \theta)) = \mathbb{E}\left(\frac{1}{1 + \exp\left(-\frac{U-\theta}{\tau}\right)}\right) - \mathbb{E}\left(\frac{1}{\exp\left(\frac{U-\theta}{\tau}\right)+1}\right) = 2\tau \log\left(\frac{\exp(\frac{1-\theta}{\tau})+1}{\exp(\frac{-\theta}{\tau})+1}\right) - 1,$$

$$\nabla_\theta \mathbb{E}(h_\tau(U, \theta)) = 2\left(\frac{1}{1 + \exp\left(\frac{1-\theta}{\tau}\right)} - \frac{1}{1 + \exp\left(\frac{-\theta}{\tau}\right)}\right).$$

Therefore,

$$[\text{Bias}(\nabla_\theta h_\tau(U, \theta))]^2 = 4 \left( 1 + \frac{1}{1 + \exp\left(\frac{1-\theta}{\tau}\right)} - \frac{1}{1 + \exp\left(\frac{-\theta}{\tau}\right)} \right)^2. \tag{11}$$

Note that the above term goes to 4 as $\tau \to \infty$. The other part contributing to the mse (10) is the variance term and it satisfies

$$\text{Var}(\nabla_\theta h_\tau(U, \theta)) \leq \mathbb{E}\left[\nabla_\theta(h_\tau(U, \theta))\right]^2 = \frac{4}{\tau} \left[ \frac{3 \exp\left(\frac{-\theta}{\tau}\right) + 1}{6 \left(\exp\left(\frac{-\theta}{\tau}\right) + 1\right)^3} - \frac{3 \exp\left(\frac{1-\theta}{\tau}\right) + 1}{6 \left(\exp\left(\frac{1-\theta}{\tau}\right) + 1\right)^3} \right].$$

Combining the above equation with (11), we arrive at $\text{MSE}(\hat{\nabla}_{\tilde{\tau},N}^{\text{grad}}) \leq v_N(\tau)$, where

$$v_N(\tau) \triangleq 4 \left( 1 + \frac{1}{1 + \exp\left(\frac{1-\theta}{\tau}\right)} - \frac{1}{1 + \exp\left(\frac{-\theta}{\tau}\right)} \right)^2 + \frac{1}{N} \frac{4}{\tau} \left[ \frac{3 \exp\left(\frac{-\theta}{\tau}\right) + 1}{6 \left(\exp\left(\frac{-\theta}{\tau}\right) + 1\right)^3} - \frac{3 \exp\left(\frac{1-\theta}{\tau}\right) + 1}{6 \left(\exp\left(\frac{1-\theta}{\tau}\right) + 1\right)^3} \right]$$

For all $N \geq 1$, we can verify that $\lim_{\tau \to \infty} v_N(\tau) = 4$ and for large enough $\tau$, $v_N(\tau)$ is increasing in $\tau$. Thus, there exists some $\tilde{\tau} > 0$ such that $\text{MSE}(\hat{\nabla}_{\tilde{\tau},N}^{\text{grad}}) < 4$. Comparing this with (9) for $\text{MSE}(\hat{\nabla}_N^{\text{RF}})$ completes the proof.

