# OpenReview forum: "Adaptive Labeling for Efficient Out-of-distribution Model Evaluation"
_NeurIPS.cc/2024/Conference — NeurIPS 2024 poster_

### Official Review · Reviewer_PJGF · 2024-06-17

**Soundness:** 3
**Presentation:** 3
**Contribution:** 2
**Rating:** 6
**Confidence:** 2

**Summary:**

This paper studies model evaluation under distribution shifts. It formulates a planning problem to select inputs for labeling from a large pool of unlabeled data. The paper then proposes an auto-differentiable planning framework to compute reliable policy gradients and update the sampling policy accordingly. To demonstrate the utility of this framework, the paper considers one-step lookahead policy based on their framework and shows that it outperforms active learning-based heuristics and REINFORCE policy gradients.

**Strengths:**

- **Clear presentation**: The paper is well-structured, with figures (e.g., Figure 2 and Figure 3) illustrating the adaptive sampling framework, which helps in understanding the proposed method.
- **Detailed explanations**: The authors provide detailed explanations of the UQ module in Section 3.1, as well as Soft K-subset sampling and Smoothing UQ module in Section 3.3. Even for readers unfamiliar with these terms, the explanations are clear enough to convey their meanings.

**Weaknesses:**

- **Computational inefficiency**: Their method requires significant computational resources, especially when the posteriors aren’t in closed form. Due to these computational constraints, Section 4.2 only considers a simple toy setting and does not compare the results with active learning and REINFORCE policy gradients. Additionally, the paper only implements one-step lookahead based on their framework, leaving the question of how to efficiently conduct multi-step planning methods unresolved.
- **Fixed and equal labeling cost assumption**: The paper assumes that the modeler pays a fixed and equal cost for each label. It would be more interesting to consider variable labeling costs, which are more common in practice.
- **Lack of theoretical guarantees**: The paper does not provide any theoretical guarantees for their method.

**Questions:**

- Why not evaluate how close $\mu_T$ is to the ground truth $f^*$? Is $\mu_T$ always guaranteed to be a distribution concentrated around $f^*$?
- How should $n$ be chosen in Algorithm 3? How should the sampling size $K\$ be selected, and how do different choices affect the results?

**Limitations:**

See Weaknesses

---

> ### Author Rebuttal · Authors · 2024-08-06
>
> We thank the reviewer for the thorough review and valuable feedback on our manuscript. Please refer to the global response for our detailed answers to the concerns raised and the changes we intend to make in the camera-ready version based on all the reviews. We also address the concerns raised point by point here.
>
>
> **Computational inefficiency: Their method requires significant computational resources, especially when the posteriors aren’t in closed form. Due to these computational constraints, Section 4.2 only considers a simple toy setting and does not compare the results with active learning and REINFORCE policy gradients. Additionally, the paper only implements one-step lookahead based on their framework, leaving the question of how to efficiently conduct multi-step planning methods unresolved.**
>
> Our work introduces a novel adaptive labeling formulation for evaluating model performance when existing labels suffer severe selection bias. As we highlight in Section 2, the adaptive labeling MDP is hard to solve because of continuous states and combinatorial actions. We make initial methodological progress by designing a one-step lookahead planning algorithm and demonstrate that even this approach can outperform existing active learning heuristics. However, this improvement comes at the cost of additional computational resources. Our formulation is thus appropriate in high-stake settings such as clinical trials where labeling costs far exceed computational costs, and we hope our new formulation spurs subsequent methodological innovations for multi-step lookaheads.
>
> We conduct an additional experiment for planning with Ensemble+ as the UQ module to compare the performance of our algorithm with other active learning based heuristics. This comparison is conducted in a more complex setting than previously discussed in the paper. The results can be found in the attached PDF (included with global response). As shown, Autodiff 1-lookahead and REINFORCE outperforms the active learning heuristic, while the performance of the Autodiff 1-lookahead and REINFORCE is similar. Based on our extensive GP experiments in Section 4.1, we anticipate that our algorithm will perform better than REINFORCE in more complex scenarios for Ensemble $+$. This presents an interesting research question for future exploration.
>
>
> **Fixed and equal labeling cost assumption: The paper assumes that the modeler pays a fixed and equal cost for each label. It would be more interesting to consider variable labeling costs, which are more common in practice.**
>
> Our framework can seamlessly accommodate variable labeling costs. Specifically, we define a cost function $c$ applied on the selected subsets $\mathcal{D}^{1:T}$ and update the objective accordingly to have a term $\lambda c(\mathcal{D}^{1:T})$ where $\lambda$ is the penalty factor that controls the trade-off between minimizing variance and cost of acquiring samples.
>
> **Lack of theoretical guarantees: The paper does not provide any theoretical guarantees for their method.**
>
> We provide a theoretical explanation of why pathwise gradients outperform REINFORCE. In a tractable example we detail in the global response, we prove pathwise gradients can achieve a lower MSE than REINFORCE.
>
> **Why not evaluate how close $\mu_T$ is to the ground truth ${f^{*}}$?**
>
> Note that the modeler never has access to the ground truth $f^*$. Similar to supervised learning losses that use $Y$ as a proxy of $f^*(X)$, our objectives considers the modeler’s belief over $f^*$ based on $Y$’s observed so far.
>
> Even in our experiments (Section 4) the $f^*$ is not known in the closed form—it is only known through an evaluation dataset generated by it and it is difficult to compare $\mu_T$ (a distribution over functions $f$) to  $f^*$. In addition to comparing the variance of the L2 loss, we also show the difference between L2 loss estimated under $\mu_T$ to the true L2 loss estimated under $f^*$ (through the available evaluation set)  in Figures 5 and 7.
>
> **Is $\mu_T$ always guaranteed to be a distribution concentrated around  ${f^{*}}$?**
>
> If our model is well-specified, that is, the prior $\mu_0$ contains $f^*$, then we are asymptotically guaranteed to concentrate around $f^*$ as we collect more samples. This is due to the well known results from Doob [1]. Empirically, this means that if the ML uncertainty quantification methodology has enough capacity, $\mu_T$ will converge to $f^*$ asymptotically. We will highlight this point in our revision.
>
>
> **How should $n$ be chosen in Algorithm 3? How should the sampling size $K$ be selected, and how do different choices affect the results?**
>
> Here $n$ is simply the number of samples in the pool set we want to query labels from. The size $K$ is assumed to be a given that is input from the analyst based on acquisition cost and labeling budget.
>
> Reference:
> [1]  Doob, J. L. (1949). Application of the theory of martingales. In Actes du Colloque
> International Le Calcul des Probabilités et ses applications (Lyon, 28 Juin – 3
> Juillet, 1948), pages 23–27

---

> > ### Comment · Reviewer_PJGF · 2024-08-07
> >
> > Thank you for your response to my comments. I have read your rebuttal and will keep my score.

---

### Official Review · Reviewer_4yoX · 2024-07-12

**Soundness:** 3
**Presentation:** 2
**Contribution:** 3
**Rating:** 6
**Confidence:** 3

**Summary:**

This work formulates the process of data labeling as a Markov Decision Process (MDP). It introduces three key components: K-subset sampling, an Uncertainty Quantification (UQ) module, and a differentiable simulator. These components enable the application of the REINFORCE policy gradient algorithm to solve the MDP. The proposed method demonstrates significant performance improvements on both synthetic and real-world data.

**Strengths:**

1. The problem of data labeling is crucial in the field of machine learning. This work formulates the data labeling challenge as a Markov Decision Process (MDP) and addresses it using a Reinforcement Learning (RL) algorithm, providing valuable insights.

2. This study adapts three key components to make the REINFORCE algorithm suitable for this problem.

3. The proposed method achieves significant performance improvements on both synthetic and real-world data.

**Weaknesses:**

The organization of Section 4 could be improved. I suggest introducing the overall description of the proposed method and the pseudo-code for Algorithm 1 after the introduction of the three key components.

**Questions:**

I have no further questions. Please refer to the weaknesses mentioned above.

**Limitations:**

The authors have discussed the limitations.

---

> ### Author Rebuttal · Authors · 2024-08-06
>
> We thank the reviewer for the thorough review and valuable feedback on our manuscript. Below, we address the specific questions raised by the reviewer. Additionally, please refer to the global response for the changes we intend to make in the camera-ready version based on all the reviews.
>
> **The organization of Section 4 could be improved. I suggest introducing the overall description of the proposed method and the pseudo-code for Algorithm 1 after the introduction of the three key components.**
>
> We thank the reviewer for their advice on improving the presentation of the paper. As mentioned in our global response, we will incorporate all their suggestions in the final version of the paper.

---

> ### Comment · Reviewer_4yoX · 2024-08-12
>
> Thank you for addressing my questions. I appreciate the effort put into clarifying my concerns and keep my score.

---

### Official Review · Reviewer_CyPE · 2024-07-13

**Soundness:** 3
**Presentation:** 2
**Contribution:** 3
**Rating:** 6
**Confidence:** 2

**Summary:**

This paper focuses on sampling efforts on out-of-distribution (OOD) examples to minimize uncertainty in the prediction model's performance over T periods. The main idea is based on a novel view of MDP, which considers beliefs as states and samples as actions. The authors also demonstrate how to make the entire pipeline differentiable for optimization. Experiments show that the proposed framework achieves better sample efficiency in reducing variance.

**Strengths:**

- The approach of considering beliefs as states and samples as actions to minimize uncertainty in the prediction model's performance over T periods is both interesting and novel. Efficiently addressing OOD samples is also a very important problem.
- Experimental results provide significant evidence that Autodiff can efficiently reduce variance.

**Weaknesses:**

- I do not fully understand some parts:
-- Can the authors provide a more detailed formulation on: (1) how the G(u) is calculated? (2) Algorithm 2 returns \pi_{\theta}.  Does the pseudo-label-based part also update the policy? How is this updating connected to the updating with ground truth labels acquired in line 6 of Algorithm 1?

- In lines 102-108 of the related work, the authors mention several active learning methods in batched settings, highlighting that they focus on classification settings while this paper mainly focuses on regression settings. Given that these methods are not used as baselines, I wonder how different the active learning settings in classification and regression are. Can methods in classification settings be easily adapted to regression settings?

-What is the size of the training dataset?

**Questions:**

1. in Line 270, what do mean by "We just consider a naive predictor, i.e., ψ(·) ≡ 0 for simplicity."

**Limitations:**

The paper includes a section on limitations.

---

> ### Author Rebuttal · Authors · 2024-08-06
>
> We thank the reviewer for the thorough review and valuable feedback on our manuscript. Below, we address the specific questions raised by the reviewer. Additionally, please refer to the global response for the changes we intend to make in the camera-ready version based on all the reviews.
>
>
> **I do not fully understand some parts: -- Can the authors provide a more detailed formulation on: (1) how the $G(\mu)$ is calculated? (2) Algorithm 2 returns $\pi_{\theta}$. Does the pseudo-label-based part also update the policy? How is this updating connected to the updating with ground truth labels acquired in line 6 of Algorithm 1?**
>
> **(1)** As shown in line 164, given any posterior belief $\mu$, $G(\mu)$ is defined as $\mathrm{Var}_{f \sim \mu}  g(f) $. Depending on the form of $g(f)$ and the belief $\mu$, $G(\mu)$ can either be derived analytically or estimated by drawing samples $f \sim \mu$ and then calculating the empirical variance of $g(f)$ across these samples.
>
>
> **(2)** We refer the reviewer to the diagram explaining our algorithm in the attached PDF (alongside the global response).
> Broadly, at each time step $t$, the input to Algorithm 2 is the current posterior state $\mu_t$, which is used to generate pseudo labels. These pseudo labels are then used to optimize the policy $\pi_\theta$ in Algorithm 2. Any update of the UQ module/posterior state within Algorithm 2 using these pseudo labels is temporary and the posterior state is reverted back to $\mu_t$ at the end of Algorithm 2. Next, we select batch $X_{t+1}$ according to the optimized policy $\pi_\theta$ from the Algorithm 2 and obtain the true labels $Y_{t+1}$. Finally, we update the UQ module/posterior belief $\mu_t$ using $(X_{t+1},Y_{t+1})$ resulting in the new posterior state $\mu_{t+1}$ at time $t+1$.
>
> **In lines 102-108 of the related work, the authors mention several active learning methods in batched settings, highlighting that they focus on classification settings while this paper mainly focuses on regression settings. Given that these methods are not used as baselines, I wonder how different the active learning settings in classification and regression are. Can methods in classification settings be easily adapted to regression settings?**
>
> As mentioned in the global response, we refer the reviewer to  our active learning-based baselines in Section 4.1. In this section, we adapt common active learning heuristics in classification settings such as uncertainty sampling (static), uncertainty sampling (sequential), and random sampling, to regression problems we focus on.
>
> **What is the size of the training dataset?**
>
>
> We direct the reviewer to Section D for the experimental details, particularly noting the training dataset sizes mentioned on lines 485-486 and 511-512.
>
> **in Line 270, what do mean by "We just consider a naive predictor, i.e., ψ(·) ≡ 0 for simplicity."**
>
>
> Recall our objective is to evaluate the performance of the model $\psi(.)$. For simplicity, we assume a naive predictor/model which outputs $0$ for all the inputs. We want to emphasize that this problem is just as complex as having a sophisticated model $\psi(.)$, as it merely serves as an input to our algorithm through objective $G(\mu)$.

---

> > ### Comment · Reviewer_CyPE · 2024-08-11
> > **Reply to rebuttal**
> >
> > Thank you for detailed response. I will maintain my score.

---

### Official Review · Reviewer_GtV8 · 2024-07-15

**Soundness:** 3
**Presentation:** 1
**Contribution:** 2
**Rating:** 5
**Confidence:** 3

**Summary:**

This paper addresses the issue of selection bias in previously collected data when ground truth labels are expensive. It introduces a computational framework for adaptive labeling, aiming to provide cost-efficient model evaluations under severe distribution shifts. The problem is formulated as a Markov Decision Process (MDP), where states are defined by posterior beliefs about model performance. New batches of labels lead to "state transitions," resulting in sharper beliefs, with the goal of minimizing overall uncertainty on model performance. The proposed method optimizes the adaptive labeling policy using pathwise policy gradients via auto-differentiation through simulated rollouts, avoiding high-variance REINFORCE policy gradient estimators. The framework is adaptable to various uncertainty quantification methods and emphasizes the importance of planning in adaptive labeling. Empirical results on both synthetic and real datasets show that even a one-step lookahead policy significantly outperforms active learning-inspired heuristics.

**Strengths:**

To my knowledge, formulating model evaluation as RL is a novel and potentially useful idea.

**Weaknesses:**

The presentation is extremely confusing. See my questions.

**Questions:**

- Line 102: Why do you mean by "Batched settings"?
- Line 107: What are lookahead policies?
- Line 134: What's the purpose of rewriting the likelihood function $p(y|f, x)$ as $p_{\epsilon}(y - f(x))$? What does this likelihood function mean?
-  Line 160: What's "F_t-measureable"?
- Eq 2: Is uncertainty equal to Var? In the intro, you defined epistemic and aleatoric uncertainty, but what's the uncertainty you're trying to minimize in Eq2?
- Line 168: What's the definition of A?
- Line 179: Missing subject.
- Line 180: What's the definition of g(W_i, \theta)?
- Figure 4: Missing y-axis label.
- Line 285: Providing intuition of empirical results before presenting the experimental results seems weird to me. I suggest stating the intution after presenting the results.
- Figure 4 and 5: What is the difference in terms of purpose between them? What should readers takes away from the comparison on variance of mean squared loss and error of MSE on collected data? Line 314 refers to them but doesn't explain the details.
- Figure 6 and 7: Same issues with Figure 4 and 5.


Minor comments:
- The related works are a lot. Consider making it a section. It's too dense.

**Limitations:**

Yes, the limitation is discussed.

---

> ### Author Rebuttal · Authors · 2024-08-06
>
> We thank the reviewer for the thorough review and valuable feedback on our manuscript. Below, we address the specific questions raised by the reviewer. Additionally, please refer to the global response for the changes we intend to make in the camera-ready version based on all the reviews.
>
> **Line 102: Why do you mean "Batched settings"?**
>
> In a batched setting, we select a group of inputs to be labeled in each step, rather than just a single input. This approach is particularly important because, in practical scenarios, labels are typically requested in batches rather than individually. Additionally, it makes the problem more challenging by inducing a combinatorial action space. We discuss this in Section 1 (Lines 47-48) and will elaborate further on the problem setup in the camera-ready version.
>
> **Line 107: What are lookahead policies?**
>
> Lookahead policies are a class of policies for approximately solving the MDPs (Markov Decision Processes). These policies make decisions by explicitly optimizing over a shorter horizon of the MDP rather than the entire horizon. Depending on the length of this shorter horizon, they are referred to as 1-step, 2-step lookahead policies. We will include the following relevant references in the main paper.
>
> [1] D. Bertsekas and J. Tsitsiklis. Neuro-dynamic programming. Athena Scientific, 1996
>
> [2] Y. Efroni, M. Ghavamzadeh, and S. Mannor. Online planning with lookahead policies. Advances in Neural Information Processing Systems,  2020
>
> [3] Yonathan Efroni, Gal Dalal, Bruno Scherrer, and Shie Mannor. Multiple-step greedy policies in approximate and online reinforcement learning. In Advances in Neural Information Processing Systems, 2018.
>
> **Line 134: What's the purpose of rewriting the likelihood function $p(y|f, x)$ as $p_{\epsilon}(y - f(x))$? What does this likelihood function mean?**
>
> We want to highlight that the only source of randomness in the label $y$, given $f$ and $x$, is the noise $\epsilon$. Furthermore, this randomness depends solely on $y-f(x)$ and the distribution of the noise $\epsilon$.
>
> **Line 160: What's "$F_t$-measureable"?**
>
> $\mathcal{F}_t$-measureable states that the posterior state $\mu_t$ is known at time $t$. This is a formal way of stating that the posterior state depends only on the prior and data acquired up to time $t$.
>
> **Eq 2: Is uncertainty equal to Var? In the intro, you defined epistemic and aleatoric uncertainty, but what's the uncertainty you're trying to minimize in Eq2?**
>
> We aim to minimize the epistemic uncertainty, specifically by considering  the variance of $g(f)$, where $f \sim \mu_T$. This variance represents the uncertainty due to a lack of knowledge about the underlying $f$, which is the epistemic uncertainty.
>
> **Line 168: What's the definition of A?**
>
> $A$ is a random variable distributed according to $\pi(\theta)$. It is introduced to highlight the difference between the REINFORCE estimator and the pathwise gradient estimator.
>
> **Line 179: Missing subject.**
>
> Thanks for pointing this out, we will address it in the main paper.
>
> **Line 180: What's the definition of $g(W_i, \theta)$?**
>
> A random variable $A$ distributed according to $\pi_\theta$ can always be expressed as $h(W, \theta)$ for some function $h(.)$, where $W$ is distributed independent of $\theta$. We use  a smoother approximation $g(W, \theta)$ of $h(W, \theta)$ to enable differentiability with respect to $\theta$. For example, in Algorithm 2, we discuss how the standard Gumbel random variable $W$ can be used to generate a softer version $a(\theta)$ of K-subset samples.
>
> **Figure 4: Missing y-axis label.**
>
> Thank you for bringing this to our attention. We will add the y-axis labels in the paper. For your reference, we have mentioned the labels in the figure description.
>
> **Line 285: Providing intuition of empirical results before presenting the experimental results seems weird to me. I suggest stating the intuition after presenting the results.**
>
> Thank you for the suggestion. We will move the intuition after presenting results in Section 4.
>
> **Figure 4 and 5: What is the difference in terms of purpose between them? What should readers take away from the comparison on variance of mean squared loss and error of MSE on collected data? Line 314 refers to them but doesn't explain the details.**
>
> Figure 4 demonstrates that the variance of the model’s L2 loss is reduced, which is the objective of the algorithm.
> In Figure 5, besides comparing the variance of the L2 loss, we also compare the accuracy of the updated L2 loss estimation after acquiring new points. This illustrates that the objective is appropriate: by minimizing the variance, we also achieve a better evaluation of the model.
>
>
> **Figure 6 and 7: Same issues with Figure 4 and 5.**
>
> This has been addressed above.
>
> **There are many related works. Consider making it a separate section, as the current content is too dense.**
>
> Thanks for pointing this out. We will make it a separate section.

---

> > ### Comment · Reviewer_GtV8 · 2024-08-08
> >
> > Thanks for the clarification. I'm increasing my rating to "Borderline accept." I'm not comfortable with increasing the rating further since the significance of the empirical results is not clear to me (even after reading the new results).

---

### Author Rebuttal · Authors · 2024-08-06

We thank all the reviewers for the valuable feedback. We appreciate the constructive comments and will incorporate all discussions below in the camera-ready version of the paper.

**Summary of contributions:** Our work introduces a *novel problem formulation* for evaluating model performance when existing data suffer severe selection bias, where state-of-the-art methods such as importance sampling are insufficient. Specifically, we formulate the adaptive labeling problem as a Markov Decision Process (MDP) over states defined by posterior beliefs on model performance. The rest of the paper highlights the potential of our novel formulation using simple methods derived from it. As we highlight in Section 2, the adaptive labeling MDP is hard to solve because of continuous states and combinatorial actions. We design a tractable *single-step lookahead* planning algorithm and demonstrate even small lookahead *outperforms existing active learning heuristics*.

Our results underscore the benefits of formulating the adaptive labeling problem as an MDP and designing specific RL algorithms to solve it. As Reviewer PJGF correctly points out, this comes at the cost of additional computational resources. Our formulation is thus appropriate in *high-stake settings such as clinical trials* where labeling costs far exceed computational costs, and we hope our new formulation spurs subsequent methodological innovations. We thank Reviewers GtV8, CyPE, 4yoX, for recognizing the unusual yet exciting dimension of having the formulation as our main contribution.

To implement one-step lookahead, we move away from standard REINFORCE policy gradients that suffer high variance and propose a novel pathwise gradient based on auto-differentiation. As we summarize in the attached PDF, our approach smoothes the non-differentiable operations to enable back propagation. Despite the bias incurred by smoothing, we observe pathwise policy gradients outperform REINFORCE buoyed by low variance (Section 4). While a substantive methodological contribution, the computational efficiency of our algorithm is bottlenecked by that of the auto-differentiation framework and we highlight promising future directions of research in the Section 4.

**Comparison with active learning:**  Reviewer CyPE asks for comparison with active learning heuristics, and we would like to point to Section 4.1 where we compared our algorithm to common active learning heuristics adapted from classification to regression settings, such as uncertainty sampling (static), uncertainty sampling (sequential), and random sampling.

**Variable labeling cost:** Our framework can seamlessly accommodate variable labeling cost. Specifically, we define a cost function $c$ applied on the selected subsets $\mathcal{D}^{1:T}$ and update the objective accordingly to have a term $\lambda c(\mathcal{D}^{1:T})$ where $\lambda$ is the penalty factor that controls the trade-off between minimizing variance and cost of acquiring samples.

**Experiments:** We have conducted an additional experiment as suggested by Reviewer PJGF. Specifically, for planning with Ensemble+ as the UQ module, we compare the performance of our algorithm with other active learning based heuristics. This comparison is conducted in a more complex setting than previously discussed in the paper. Please refer to the attached PDF for the results. Based on the extensive GP experiments in Section 4, we expect that our results will extend to even more complex scenarios for Ensemble $+$.

**Theoretical insights:** Incorporating Reviewer PJGF’s suggestion, we provide a theoretical explanation of why pathwise gradients can outperform REINFORCE, despite being biased due to smoothing. Recall that in our adaptive labeling framework, we deal with a combinatorial action space where actions $A\in\mathcal{A} = \lbrace a_1,...,a_m \rbrace$ are drawn according to a policy $\pi_\theta$. The objective is to minimize $G(\theta)=E_{A\sim\pi_\theta}f(A)$, which requires estimating $\nabla_\theta G(\theta)$. We consider a tractable yet insightful setting where $\mathcal{A} =  \lbrace -1,1  \rbrace$ with $\pi_\theta(-1)=\theta$ and $f(A)=A$.


The REINFORCE estimator $\nabla_N^{RF}$ is a $N$-sample approximation of $E_{A\sim\pi_\theta}[f(A)\nabla_\theta\log(\pi_\theta(A))]$. Note that $G(\theta)$ can be rewritten as $E_Uf(h(U,\theta))$, where $h(U,\theta):=2I(U > \theta)-1\stackrel{d}{=}A$ and $U\sim[0,1]$. To enable the pathwise policy gradient, we consider a smoothed $h^{\tau}(U,\theta)$ using sigmoid approximation, where $\tau$ is the temperature. Pathwise gradient estimator $\nabla_{\tau,N}^{grad}$ is the $N$-sample approximation of $E_U[\nabla_\theta f(h^{\tau}(U,\theta))]$.

**Theorem:** For $\theta\in (0,1)$ and $N<\frac{1}{4\theta(1-\theta)}$, there exists $\tau$, depending on $N,\theta$ such that $E[mse(\nabla_{\tau,N}^{grad})]<4<E[mse(\nabla_N^{RF})]$. Further, for any $N$, as $\theta=\frac{1}{k}$ and $k\to\infty$, $E[mse(\nabla_N^{RF})]=\Omega(k)$, while $E[mse(\nabla_{\tau,N}^{grad})]<4.$ The same statement holds for $\theta=1 - \frac{1}{k}$.


**Presentation:** We have taken to heart reviewer comments on improving our presentation. We outline a roadmap for enhancing the presentation of the paper below:

(1) Section 1: As suggested by reviewer GtV8, we will create a separate section for related works.

(2) Section 3: We will first explain all the major components of our algorithm before introducing Algorithm 1 and 2. Additionally, we plan to include the figure (from the attached PDF) in the main paper.

(3) Section 4: We will reorganize this section and will provide intuition after we present our experimental results, as suggested by Reviewer GtV8. In addition, we will make active learning based heuristics a separate subsection to convey the results better.

---

### Decision · Program_Chairs · 2024-09-25

**Decision:**

Accept (poster)

**Comment:**

The paper studies the problem of model evaluation under distribution shift. It formulates adaptive labeling as a planning problem to select inputs for labeling from a large pool of unlabeled samples. To show the efficacy of this approach, it considers one-step look-ahead policy and shows that it outperforms several heuristics based on active learning and REINFORCE policy gradient method. Although the authors found the problem studied in the paper interesting and the approach novel, there are serious concerns about the clarity of presentation, the computational inefficiency of the proposed approach, and the lack of theoretical guarantees. Despite all the above, they still found the paper (slightly) above the bar. I would recommend the authors to revise the paper, significantly improve its presentation and properly address the comments/questions raised by the reviewers (to include some of the responses during the rebuttals in the paper) for the next revision.